# Effect of Rainfall on Soil Aggregate Breakdown and Transportation on Cultivated Land in the Black Soil Region of Northeast China

**Yikai Zhao [1], Han Wang [1], Xiangwei Chen [1,2,\*] and Yu Fu [1,\*]**

[1] School of Forestry, Northeast Forestry University, Harbin 150040, China
[2] Key Laboratory of Sustainable Forest Ecosystem Management Ministry of Education, Northeast Forestry University, Harbin 150040, China
\* Correspondence: chenxwnefu@163.com (X.C.); aily_fy@163.com (Y.F.)

**Abstract:** To clarify the characteristics of soil aggregate breakdown and transportation by rainfall, the cultivated land (0–10 cm) of the black soil region of Northeast China was taken as the research object, with rainfall intensities of 78 and 127 mm·h$^{-1}$ ($RI_{78}$, $RI_{127}$), to analyze the differences in the mass transported, fraction size distribution, mean weight diameter (*MWD*) and enrichment rate of aggregates before and after runoff generation. Before runoff generation, the total mass of transported aggregates, the total mass percentage of the aggregate fraction size < 1 mm and the *MWD* of the transported aggregates were not significantly different at rainfall intensities of 78 and 127 mm·h$^{-1}$. After runoff generation, the mass of transported aggregates was greater than that before runoff generation, and the mass of transported aggregates under $RI_{78}$ was significantly higher than that under $RI_{127}$, by 15.6%. The mass percentage of the aggregate fraction size < 0.053 mm after runoff generation was significantly reduced by 8.4% and 19.4% compared to that before runoff generation. After runoff generation, compared with before runoff generation, the *MWD* of the transported aggregates was significantly reduced by 50.9% and 50.3% under $RI_{78}$ and $RI_{127}$, respectively. Compared with before runoff generation, the mass percentage of small macroaggregates increased gradually with the increase in the transport distance. The aggregate fraction size > 0.25 mm was lost, and the fraction size < 0.25 mm was enriched, before and after runoff generation. A comparative study on the characteristics of black soil aggregate breakdown and transportation before and after runoff generation can provide a theoretical basis for the mechanism of soil erosion and the transportation of cultivated soil in the black soil region of Northeast China.

**Keywords:** aggregates; fraction size distribution; mean weight diameter; transport distance

## 1. Introduction

Soil aggregates are an important part of the soil structure and are the basic unit of soil resistance to external forces [1,2]. Aggregate breakdown caused by rainfall is a critical part of soil erosion [3,4]. On the one hand, raindrops break up soil aggregates, although raindrops also have a sorting effect on the aggregate fraction size [5]: macroaggregates are split, and microaggregates are transported, which results in a redistribution of the aggregate size, and the destruction of the topsoil structure during rainfall. On the other hand, the broken and transported small aggregates block the surface-soil pores, form crusts and enhance the turbulence intensity of the thin-layer runoff, which results in the enhanced soil-erosion capacity by rainfall [6]. Therefore, aggregate breakdown and transportation under rainfall have an important impact on changes in the soil structure.

The study of aggregate breakdown and transportation under rainfall has mainly focused on raindrop detachment and splash transport [7–10]. Liu et al. [11] conducted simulation rainfall experiments (rainfall intensity: 90 mm·h$^{-1}$; rainfall duration: 45 min) in a cylindrical container with drainage holes, and the splash erosion was relatively stable

in the later period of rainfall. Ma et al. [12] found that the total mass of the splash erosion increased as a power function (rainfall intensity: 58 mm·h$^{-1}$; rainfall duration: 61 min). This was because the test device used by Ma et al. [12] could not discharge hydrops, and hydrops carried loose soil particles into the collection device. Therefore, the mass of splashes would increase with the increasing rainfall duration. Legout et al. [13] found that the splash erosion mass and mean weight diameter (*MWD*) of silty clay loam were the maximum only at a near transport distance. However, most researchers believe that the higher the *MWD* of splash erosion aggregates, the smaller the splash erosion amount should be, which is contrary to the above [14–16]. This difference may be because, under the experimental conditions of Legout et al. [13], the rainfall duration was long, and the infiltration rate was smaller than the rainfall intensity, which easily resulted in hydrop accumulation. The accumulated hydrop layer will affect the impact of raindrops [17]. Based on the above studies, the surface-soil hydrops generated by long rainfall duration and rainfall conditions under high rainfall intensity are important factors that affect the test results of the splash erosion process. However, most studies did not consider the impact of hydrop accumulation on aggregate breakdown and transportation. This may be due to the following two reasons: first, the breakdown and transportation of aggregates are less affected by hydrop accumulation, which is usually ignored; second, the test device for studying splash erosion is usually placed horizontally, and so it is impossible to discharge more water, except through soil infiltration.

The effect of raindrops on the breakdown and transportation of soil aggregates did not disappear after the formation of thin-layer runoff. The macroaggregates were deposited on the downhill slope under the combined action of raindrops and runoff, and microaggregates were transported farther with runoff generation [18]. During the whole rainfall process, Zhang et al. [19] found that the soil-erosion rate decreased rapidly before runoff generation, and then decreased slowly until it stabilized after runoff generation. Mahmoodabadi and Sajjadi [20] showed that the erosion loss after runoff generation was much greater than the erosion loss before runoff generation. However, Van et al. [21] considered that the erosion loss after runoff generation was 8–22% of the erosion loss before runoff generation. Moreover, Ghahramani et al. [22] believed that the erosion loss before runoff generation should be multiplied by the erosion loss after runoff generation. Hu et al. [23] discharged hydrops by adjusting the slope of the test device (10°) in the simulated rainfall test, but they did not distinguish and compare the transported aggregates before and after runoff generation.

The black soil region of Northeast China is an important commodity grain production base [24]. Water erosion is the main type of soil erosion in this area [25], and it mainly occurs on sloping farmland [26]. Moreover, the contribution of rainfall to cultivated-land erosion exceeds 78.3% [11]. At present, cultivated cropland has the greatest soil-erosion rate [27], which leads to the destruction of the soil structure and the loss of soil nutrients, resulting in a decline in the soil quality and a weakening of the fertilizer supply capacity. As the basic unit of the soil structure and a carrier of soil nutrients, soil aggregates play an important role in maintaining the soil quality and undertaking nutrient transformation and transportation. The characteristics of aggregate breakdown and transportation before runoff generation in black soil regions have been studied [23,28]. However, less attention has been given to the changes in aggregate breakdown and transportation after runoff generation, which cannot reveal the characteristics of soil aggregate breakdown and transportation driven by rainfall. Therefore, the study of aggregate breakdown and transportation before and after runoff generation includes the following three aspects: (i) clarifying the impact of rainfall on the fraction size distribution of aggregates; (ii) analyzing the change in the mass of aggregates transported; (iii) revealing the variation in the characteristics of aggregates in different fraction sizes with the transport distance. It is of great significance to study the aggregate breakdown and transportation characteristics before and after runoff generation to further understand the water erosion mechanism.

## 2. Materials and Methods

### 2.1. Soils

The sampling site is located in Binxian County (126°55′–128°19′ E, 45°30′–46°01′ N), Harbin city, Heilongjiang Province, China. The slope variation is mainly between 1 and 8°. The thickness of the black soil layer is generally less than 30 cm, making it a typical thin black soil region (Mollisols, USDA Soil Taxonomy System). The average annual precipitation is 681 mm. Maize is the main cultivated crop. Soil samples were collected before sowing. Six undisturbed soil samples were collected by a rectangular sampler (length 40 cm × height 10 cm × width 10 cm). The soil bulk density at the sampling point was determined by the cutting ring method, and three soil samples were taken for the determination of organic matter, pH and soil mechanical fraction size distribution. The soil bulk density $(1.2 \pm 0.1)$ g·cm$^{-3}$, organic matter $(33.7\% \pm 3.8\%)$, soil pH $(5.5 \pm 0.1)$, soil moisture content $(27.0\% \pm 0.3\%)$, sand particle content $(27.2\% \pm 3.9\%)$, silt particle content $(41.5\% \pm 3.6\%)$ and clay particle content $(31.3\% \pm 3.4\%)$ of the test soil samples were measured.

### 2.2. Rainfall Simulation

#### 2.2.1. Test Device

The test device consisted of three parts: a rainfall device, a water supply tank and an aggregate transported collection device (Figure 1). The rain collector consisted of a homemade needle-type stainless steel rain collector (length 40 cm × height 20 cm × width 10 cm). The water level of the water supply tank corresponding to the rainfall intensity selected for this test level of continuous water supply was measured. The support frame could adjust the slope (0–10°). The test soil tank containing undisturbed soil had the following dimensions: length 40 cm × height 10 cm × width 10 cm. The aggregate transport collection device (length 140 cm × width 110 cm) was improved based on Ellison [29]. Drainage holes with a radius of 2 cm were arranged in the same direction of each transport distance (for collecting transported aggregates). The outlet trapezoidal groove was connected to the top of the test soil tank and was used only to discharge rainfall runoff. Baffles were arranged every 10 cm from the edge of the test soil bin (each migration distance was 1 cm above the baffle). The runoff discharge outlet and splash plate did not affect each other, and the aggregate transport collection device was closely combined with the test soil bin to make it impervious to water. The rainfall height was 2 m.

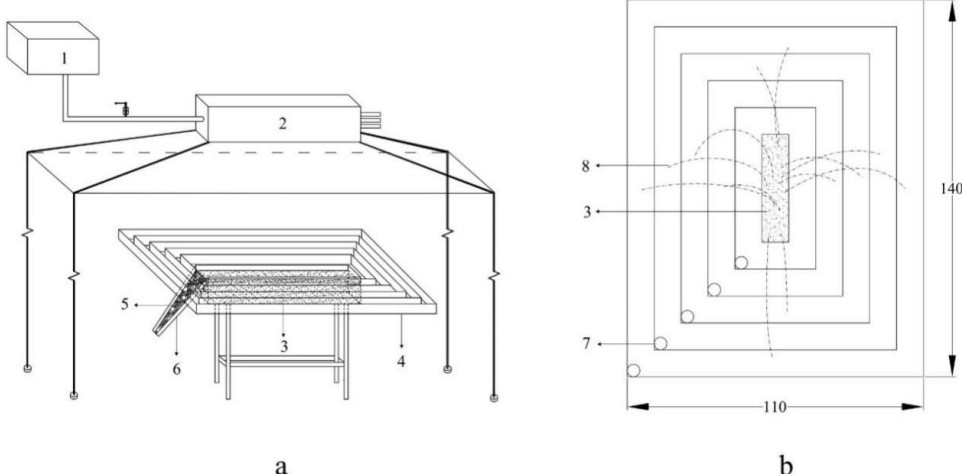

**Figure 1.** Test device. (**a**) test device diagram. (**b**) schematic diagram of the splash plate. Note: 1: water supply tank. 2: rain collector. 3: test soil tank. 4: splash plate. 5: runoff outlet—the trapezoidal groove. 6: baffle. 7: drain hole. 8: splashed soil particles at different transport distances.

### 2.2.2. Experimental Design

The average values of July (120–150 mm), with the most abundant rainfall in the black soil region, and June and August (60–120 mm), with the second highest rainfall amounts, were selected as the basis for the selection of rainfall intensity. After multiple calibrations, the rainfall intensity of this experiment was designed to be 78 and 127 mm·h$^{-1}$. In the black soil region of Northeast China, the rainfall duration is mostly less than 1 h [30]. Li et al. [10] studied the characteristics of soil and water loss and found that the area with a slope < 6° exceeded 97% in the black soil region of Northeast China. Therefore, a slope of 4° was selected. There were 3 replicates for each experiment. The main parameters of rainfall and runoff are shown in Table 1.

**Table 1.** Main parameters of rainfall and runoff.

| Experimental Treatment | Raindrop Diameter (mm) | Average Velocity (cm·s$^{-1}$) | Average Runoff Depth (mm) |
|:---:|:---:|:---:|:---:|
| $RI_{78b}$ | 3.46 | | |
| $RI_{78a}$ | 3.46 | 0.88 | 0.73 |
| $RI_{127b}$ | 3.66 | | |
| $RI_{127a}$ | 3.66 | 1.27 | 0.82 |

Note: $RI_{78b}$ and $RI_{78a}$ represent the test results before and after runoff generation under a rainfall intensity of 78 mm·h$^{-1}$; $RI_{127b}$ and $RI_{127a}$ represent the test results before and after runoff generation under a rainfall intensity of 127 mm·h$^{-1}$.

Before the experiment, the cuboid sampler containing the undisturbed soil was saturated for 12 h in a container with a water level that was 1 cm below the topsoil surface to ensure that the original moisture content of each sample was consistent. The rainfall intensity was calibrated to meet the test standard, and the test was started and timed. During the experiment, when there was runoff outflow at the outlet of runoff discharge, the time of runoff generation was recorded. The total mass of transported aggregates before runoff generation was collected separately. The rainfall duration was 30 min after runoff generation. The transported aggregates after runoff generation were collected every 5 min. During this period, a KMnO$_4$ solution and stopwatch were used to measure the runoff flow rate, and the runoff depth was calculated simultaneously. The sum collected within 30 min was the total mass of transported aggregates after runoff generation. This test was used only to compare the total mass of transported aggregates before and after runoff generation. After the experiment, the collected aggregates were passed through 5, 1, 0.25 and 0.053 mm sieves in turn, and five fraction sizes (>5, 5–1, 1–0.25, 0.25–0.053 and <0.053 mm) were obtained using an aggregate analyzer (HR-TTF-100). All aggregates were oven-dried for 24 h at 40 °C and then weighed.

### 2.3. Index Calculation Methods

In the rainfall process, the runoff flow rate measured by the KMnO$_4$ dye method is the maximum velocity on the slope, which needs to be multiplied by the correction coefficient as the average runoff velocity on the slope. The expression is:

$$V = kV_m \tag{1}$$

where k is the correction factor. k is 0.67 in laminar flow, 0.7 in transitional flow and 0.8 in turbulent flow [31]. The Reynolds number (*Re*) can be used to determine whether the flow is laminar or turbulent. When *Re* > 800, the flow is turbulent. When *Re* < 800, the flow is laminar [32]. The correction factor k was 0.75 in this study.

It was assumed that the runoff was uniformly distributed along the slope. The runoff depth was calculated as follows:

$$h = \frac{q}{v\,b\,t} \tag{2}$$

where $h$ is the runoff depth, m; $q$ is the runoff at time $t$, m$^3$, determined by the sediment sample of the runoff; $v$ is the runoff velocity of the cross section, m/s, determined by the staining method; and $b$ is the width of the cross section, measured by a thin steel ruler, m.

The mass percentage of aggregates refers to the percentage of the mass of aggregates with a certain fraction size in the total mass. It is often used to indicate the fraction size distribution of aggregates [12,13].

$$M = \frac{m_1}{m_2} \tag{3}$$

where $M$ is the mass percentage of aggregates, %; $m_1$ is the mass of aggregates with a certain fraction size, mg; $m_2$ is the total mass of aggregates, mg.

The *MWD* of soil aggregates can be used as one of the evaluation indexes of aggregate distribution. The higher the *MWD*, the higher the content of large-fraction aggregates after crushing. This study used the method in Le Bissonnais [33] to calculate *MWD*:

$$E_{rain} = \frac{1}{2}mv^2 \tag{4}$$

where $r_i$ is the pore size (mm) of each sieve, $r_0 = r_1$ and $r_n = r_{n+1}$; $m_i$ is the mass percentage of aggregates in the grade $i$ sieve; and $n$ is the number of sieves.

For each size fraction, the enrichment ratio (*ER*) refers to the mass percentage of splash sediment compared to the undisturbed aggregates. An *ER* value > 1 indicates enrichment of the fraction, whereas an *ER* value < 1 indicates depletion of the fraction. The *ER* is calculated as follows:

$$ER = \frac{P_{sp}}{P_{sa}} \tag{5}$$

where $P_{sp}$ is the mass percentage of one size fraction in the total mass of splashed aggregates and $P_{sa}$ is the mass percentage of a fraction of the total primary aggregates.

### 2.4. Data Processing

SPSS 16.0 (IBM SPSS Software, Armonk, NY, USA) was used to compare the differences in the total mass percentage of transported aggregates (Table 2), the mass of transported aggregates (Table 3) and the mass percentage of aggregates before and after runoff generation at different transport distances (Table 4). The relationship between the mass of transported aggregates and transport distance was analyzed by SPSS 16.0 (Table 3), and all figures (Figures 2 and 3) were generated in Origin 8.5 (OriginLab Corporation, Northampton, MA, USA).

## 3. Results

### 3.1. Fraction Size Distribution of Aggregates

The total mass percentage of aggregates before and after runoff generation for each fraction size is shown in Table 2. Overall, the fraction size > 5 mm aggregates were not transported before and after runoff generation. The fraction size < 1 mm aggregates accounted for 96.4–99.5% of the total, of which the fraction size < 0.053 mm aggregates accounted for the highest proportion, followed by the 1–0.25 mm aggregates. The *MWD* of transported aggregates after runoff generation was lower than that before runoff generation.

Compared with $RI_{127b}$, the total mass percentage of the 5–1 mm aggregates was significantly increased by 98.3% under $RI_{78b}$ ($p > 0.05$), and the total mass percentage of the <1 mm aggregates was not significantly different under $RI_{78b}$ ($p > 0.05$). The *MWD* was not significantly different between $RI_{78b}$ and $RI_{127b}$ ($p > 0.05$). Compared with $RI_{127a}$, the total mass percentage of 1–0.25 mm aggregates was significantly increased by 30.8% under $RI_{78a}$ ($p < 0.05$). Compared with $RI_{127a}$, the *MWD* increased by 36.0% under $RI_{78a}$ ($p < 0.05$).

Compared with $RI_{78b}$ and $RI_{127b}$, the total mass percentage of 5–1 mm and <0.053 mm aggregates significantly decreased by 77.3% and 77.7% and 8.4% and 19.4% under $RI_{78a}$ and $RI_{127a}$ ($p < 0.05$), respectively, and the total mass percentage of 1–0.25 mm aggregates increased by 35.9% under $RI_{78a}$ ($p < 0.05$). Compared with $RI_{78b}$ and $RI_{127b}$, the total

mass percentage of 0.25–0.053 mm aggregates increased by 18.9% under $RI_{127a}$ ($p < 0.05$). Compared with $RI_{78b}$ and $RI_{127b}$, the *MWD* of aggregates significantly decreased by 50.9% and 53.0%, respectively, under $RI_{78a}$ and $RI_{127a}$ ($p < 0.05$).

**Table 2.** Total mass percentage of differently sized aggregates before and after runoff generation.

| Experimental Treatment | Total Mass Percentage of Aggregates Under Experimental Treatment (mg) | | | | *MWD* (μm) |
|---|---|---|---|---|---|
| | 5–1 mm | 1–0.25 mm | 0.25–0.053 mm | <0.053 mm | |
| $RI_{78b}$ | 4.6 [Aa] | 25.9 [Aa] | 18.0 [Aa] | 51.5 [Aa] | 258 [Aa] |
| $RI_{78a}$ | 1.1 [Ba] | 35.2 [Ba] | 22.2 [Aa] | 41.5 [Ba] | 127 [Ba] |
| $RI_{127b}$ | 2.3 [Ab] | 23.4 [Aa] | 16.4 [Aa] | 57.9 [Aa] | 198 [Aa] |
| $RI_{127a}$ | 0.5 [Ba] | 26.9 [Ab] | 19.5 [Ba] | 53.0 [Ba] | 93 [Ba] |

Note: $RI_{78b}$ and $RI_{78a}$ represent the test results before and after runoff generation under a rainfall intensity of 78 mm·h$^{-1}$; $RI_{127b}$ and $RI_{127a}$ represent the test results before and after runoff generation under a rainfall intensity of 127 mm·h$^{-1}$. Different capital letters (A, B) indicate a significant difference before and after runoff generation under the same rainfall intensity ($p < 0.05$); different small letters (a, b) indicate a significant difference before and after runoff generation at different rainfall intensities ($p < 0.05$).

### 3.2. Aggregate Enrichment Rate (ER)

The change in *ER* values before and after runoff generation is shown in Figure 2. The *ER* values of the >0.25 mm aggregates were less than 1 before runoff generation. The *ER* values of the <0.25 mm aggregates were higher than 1 before runoff generation. Compared with $RI_{78b}$, the *ER* values of the 5–1 and <0.053 mm aggregates were higher under $RI_{127b}$, and the *ER* values of 1–0.25 and 0.25–0.053 mm aggregates were lower under $RI_{127b}$.

The *ER* values of the >0.25 mm aggregates were less than 1 except under $RI_{78a}$. Compared with $RI_{78a}$, the *ER* value of the <0.053 mm aggregates was higher in $RI_{127a}$, but the *ER* value of the other aggregate fraction sizes was the opposite.

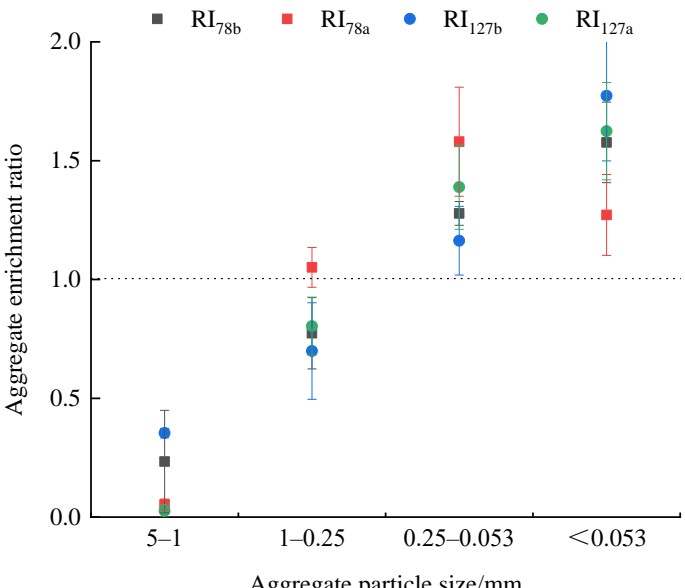

**Figure 2.** Change in the aggregate enrichment rate before and after runoff generation. Note: $RI_{78b}$ and $RI_{78a}$ represent the test results before and after runoff generation under a rainfall intensity of 78 mm·h$^{-1}$; $RI_{127b}$ and $RI_{127a}$ represent the test results before and after runoff generation under a rainfall intensity of 127 mm·h$^{-1}$.

### 3.3. Transported Characteristics of Aggregates

The mass of transported aggregates before and after runoff generation is shown in Table 3. Overall, there was no significant difference in the total mass of transported aggregates under $RI_{78}$ and $RI_{127}$ ($p > 0.05$).

The total mass of transported aggregates under $RI_{78b}$ increased by 4.3% compared with that under $RI_{127b}$ before runoff generation (Table 3). The mass of transported aggregates ($M$) after runoff generation decreased (power function relationship) with increasing runoff generation time ($T$) ($R^2 = 0.904$ and $0.922$, $p < 0.05$, $n = 6$). With the extension of the runoff generation time, the mass of transported aggregates under $RI_{78}$ and $RI_{127}$ tended to be stable from 15 to 20 min ($p > 0.05$). The total mass of transported aggregates after runoff generation was greater than that before runoff generation, and the total mass of transported aggregates under $RI_{78a}$ was significantly higher than that under $RI_{127a}$ by 15.6% ($p < 0.05$).

**Table 3.** Mass of aggregates transported before and after runoff generation.

| Experimental Treatment | Total Mass of Aggregates Transported before Runoff Generation (mg·cm$^{-1}$) | Occurrence Time of Runoff Generation (min) | Mass of Aggregates Transported after Runoff Generation for Each Time | | | | | | Total Mass of Aggregates Transported after Runoff Generation (mg·cm$^{-1}$) | Total Mass of Aggregates Transported before and after Runoff Generation (mg·cm$^{-1}$) |
|---|---|---|---|---|---|---|---|---|---|---|
| | | | 5 min | 10 min | 15 min | 20 min | 25 min | 30 min | | |
| $RI_{78}$ | 54.9 ± 7.0 | 17 | 20.7 ± 1.4 [a] | 13.3 ± 1.6 [b] | 10.4 ± 1.7 [c] | 9.9 ± 1.9 [c] | 8.2 ± 0.6 [c] | 9.7 ± 1.0 [c] | 72.3 ± 2.0 [A] | 127.2 ± 11.7 [A] |
| $RI_{127}$ | 52.7 ± 7.4 | 7 | 17.5 ± 1.5 [a] | 12.4 ± 1.5 [b] | 9.8 ± 0.3 [c] | 7.8 ± 0.8 [cd] | 6.8 ± 0.8 [d] | 8.2 ± 0.3 [d] | 62.5 ± 3.7 [B] | 115.2 ± 8.9 [A] |

Note: $RI_{78}$ and $RI_{127}$ represent rainfall intensities of 78 and 127 mm·h$^{-1}$, respectively. Different capital letters (A, B) indicate significant differences among different rainfall intensities ($p < 0.05$); different lowercase letters (a, b, c, d) indicate a significant difference after runoff generation for each time under the same rainfall intensity ($p < 0.05$).

The distribution of the mass percentage of aggregates before and after runoff generation at each transport distance is shown in Table 4. Overall, with the increase in the transport distance, the mass percentage of the 5–1 mm aggregates fluctuated, the mass percentage of the 1–0.053 mm aggregates gradually increased, and the mass percentage of the <0.053 mm aggregates gradually decreased.

Under $RI_{78b}$ and $RI_{127b}$, when the transport distance was <40 cm, the main aggregate fraction size was <0.053 mm, and the mass percentage of aggregates accounted for 40.0–54.8% and 43.6–62.9%, respectively (Figure 3). When the transport distance was >40 cm, the main aggregate fraction size was 1–0.25 mm, and the mass percentage of aggregates accounted for 50.5% and 40.8%, respectively. Compared with $RI_{127b}$, the mass percentage of aggregates with a fraction size of 5–1 mm increased significantly by 162.8% only within the transport distance of 0–10 cm under $RI_{78b}$ ($p < 0.05$). There was no significant difference in the mass percentage of other aggregates at any transport distance (Table 4).

Under $RI_{78a}$, when the transport distance was <20 cm, the main aggregate fraction size was <0.053 mm, and the mass percentage of aggregates accounted for 43.9–54.3%. When the transport distance was >20 cm, the main aggregate fraction size was 1–0.25 mm, and the mass percentage of aggregates accounted for 40.6–47.1%. Under $RI_{127a}$, when the transport distance was <30 cm, the main aggregate fraction size was <0.053 mm, and the mass percentage of aggregates accounted for 44.6–67.7%. When the transport distance was >30 cm, the main aggregate fraction size was 1–0.25 mm, and the mass percentage of aggregates accounted for 34.9–37.6%. Compared with $RI_{127a}$, the mass percentage of 5–1 mm aggregates decreased significantly by 77.7% ($p < 0.05$) only within the transport distance of 0–10 cm under $RI_{78a}$. The mass percentage of 1–0.25 mm aggregates decreased significantly by 20.6% ($p < 0.05$) within the transport distance of 10–20 cm, and there was no significant change among the other transport distances ($p > 0.05$) under $RI_{78a}$. The mass percentage of 0.25–0.053 mm aggregates increased significantly by 38.5% only within the transport distance of 40–50 cm ($p < 0.05$) under $RI_{78a}$. The mass percentage of <0.053 mm aggregates showed no significant difference at any transport distance ($p > 0.05$) under $RI_{78a}$.

Compared with before runoff generation, the mass percentage of 5–1 mm aggregates after runoff generation increased significantly by 23.0% ($p < 0.05$) under $RI_{78a}$ when the transport distance was 0–10 cm. Under $RI_{78}$ and $RI_{127}$, the mass percentage of 1–0.25 mm

aggregates increased significantly by 58.7% and 37.4% after runoff generation ($p < 0.05$) compared to that before runoff generation within the transport distance of 10–20 cm. The mass percentage of <0.053 mm aggregates decreased significantly by 22.8% and 17.4% after runoff generation compared with that before runoff generation within the transport distance of 20–30 cm ($p < 0.05$). There was no significant change in the mass percentage of 0.25–0.053 mm aggregates under $RI_{78}$ and $RI_{127}$.

**Table 4.** Fraction size distribution of aggregates at different transport distances before and after runoff generation.

| Transport Distance (cm) | Fraction Size (mm) | Mass Percentage of Aggregates Under Experimental Treatment (%) | | | |
|---|---|---|---|---|---|
| | | $RI_{78b}$ | $RI_{78a}$ | $RI_{127b}$ | $RI_{127a}$ |
| 0–10 | 5–1 | 6.4 [Aa] | 1.8 [Ba] | 2.4 [Ab] | 0.5 [Ab] |
| | 1–0.25 | 22.4 [Aa] | 23.5 [Aa] | 20.6 [Aa] | 17.6 [Aa] |
| | 0.25–0.053 | 16.5 [Aa] | 20.3 [Aa] | 14.0 [Aa] | 14.2 [Aa] |
| | <0.053 | 54.8 [Aa] | 54.3 [Aa] | 62.9 [Aa] | 67.7 [Aa] |
| 10–20 | 5–1 | 3.1 [Aa] | 0.9 [Aa] | 1.7 [Aa] | 0.3 [Aa] |
| | 1–0.25 | 21.2 [Aa] | 33.7 [Ba] | 22.1 [Aa] | 26.7 [Ab] |
| | 0.25–0.053 | 18.9 [Aa] | 21.6 [Aa] | 19.2 [Aa] | 19.9 [Aa] |
| | <0.053 | 56.8 [Aa] | 43.9 [Ba] | 57.1 [Aa] | 53.1 [Aa] |
| 20–30 | 5–1 | 3.6 [Aa] | 0.3 [Aa] | 1.3 [Aa] | 0.1 [Aa] |
| | 1–0.25 | 28.7 [Aa] | 40.6 [Aa] | 24.7 [Aa] | 33.9 [Ba] |
| | 0.25–0.053 | 21.1 [Aa] | 24.7 [Aa] | 20.0 [Aa] | 21.4 [Aa] |
| | <0.053 | 46.6 [Aa] | 34.4 [Aa] | 54.0 [Aa] | 44.6 [Ba] |
| 30–40 | 5–1 | 2.5 [Aa] | 0.3 [Aa] | 3.1 [Aa] | 0.6 [Aa] |
| | 1–0.25 | 37.0 [Aa] | 43.8 [Aa] | 35.5 [Aa] | 34.9 [Aa] |
| | 0.25–0.053 | 20.5 [Aa] | 25.7 [Aa] | 17.8 [Aa] | 24.4 [Aa] |
| | <0.053 | 40.0 [Aa] | 30.2 [Aa] | 43.6 [Aa] | 34.0 [Aa] |
| 40–50 | 5–1 | 0.0 [Aa] | 1.2 [Aa] | 3.1 [Aa] | 1.3 [Aa] |
| | 1–0.25 | 50.5 [Aa] | 47.1 [Aa] | 40.8 [Aa] | 37.6 [Aa] |
| | 0.25–0.053 | 18.9 [Aa] | 19.8 [Aa] | 24.9 [Aa] | 27.5 [Ab] |
| | <0.053 | 30.6 [Aa] | 31.9 [Aa] | 31.2 [Aa] | 33.6 [Aa] |

Note: $RI_{78b}$ and $RI_{78a}$ represent the test results before and after runoff generation under a rainfall intensity of 78 mm·h$^{-1}$; $RI_{127b}$ and $RI_{127a}$ represent the test results before and after runoff generation under a rainfall intensity of 127 mm·h$^{-1}$. Different capital letters (A, B) indicate that the particle size distribution of aggregates before and after runoff generation under the same rainfall intensity was significantly different ($p < 0.05$); different lowercase letters (a, b) indicate that there was a significant difference between different rainfall intensities before and after runoff generation ($p < 0.05$).

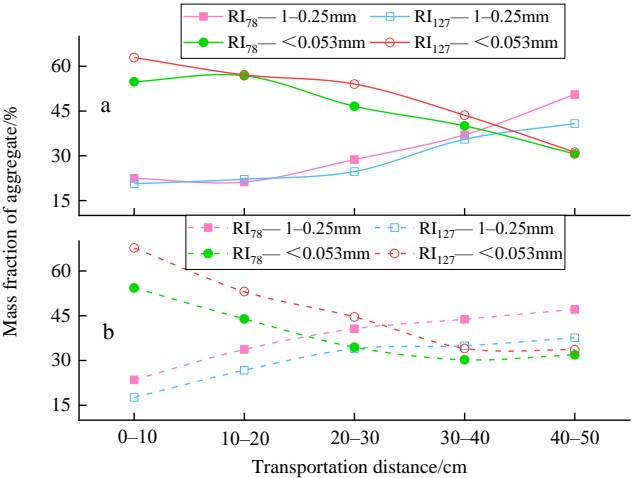

**Figure 3.** Variation in the mass percentage of 1–0.25 and <0.053 mm aggregates with transport distance. Note: (**a**) and (**b**) indicate before and after runoff generation; $RI_{78}$ and $RI_{127}$ represent rainfall intensities of 78 and 127 mm·h$^{-1}$, respectively.

## 4. Discussion

### 4.1. The Impact of Rainfall on the Fraction Size Distribution of Aggregates

At present, research on the breakdown and transportation of aggregates has mainly focused on splashing. Huang et al. [34] and Zhang et al. [35] suggested that the fraction size of aggregates transported by rainfall before runoff generation is mainly <1 mm. The result of the study also showed that the main fraction size of aggregates transported after runoff generation was <1 mm. When the aggregate fraction sizes were >1 mm, the raindrop kinetic energy was not sufficient to transport aggregates. In addition, compared with before runoff generation, after runoff generation, $RI_{78}$ significantly transported macroaggregates, and $RI_{127}$ significantly transported small macroaggregates (Table 2). The main reason was that the runoff depth increased with increasing rainfall intensity (Table 1), which weakened the impact of raindrops [17], resulting in the ability to transport small macroaggregates by RD-RIFT (raindrop detachment with transport by raindrop-induced flow transport) being different [36]. Compared with before runoff generation, the total mass percentage of <0.053 mm aggregates after runoff generation decreased significantly in the two rainfall intensity groups. Runoff transported more aggregates with fraction sizes <0.053 mm [37], which lowered the content in the loose materials of the surface soil, resulting in a decrease in the total mass percentage of <0.053 mm aggregates by splashing.

Compared with that before runoff generation, the *MWD* of aggregates decreased significantly after runoff generation (Table 1). The main reason is the effect of raindrops on runoff, which increases the carrying capacity of runoff [38,39]. Then, loose materials in the topsoil increased, which increased the mass of transported aggregates with fraction sizes of 1–0.053 mm by splashing (Table 2). Zeng et al. [16] showed that the higher the global *MWD* of splashed aggregates, the lower the splash mass. Ma et al. [12] reported that the critical size fraction was 1 mm for the enrichment and loss of Ultisol aggregates. However, the >0.25 mm aggregates were lost, and the fraction size <0.25 mm was enriched before and after runoff generation (Figure 2). This was mainly due to the following reasons: first, undisturbed soil was selected in this study, but Ma et al. [12] selected oven-dried aggregates of 5–2 mm as the research object; second, the diameter of raindrops in this study was 3.36 and 3.66 mm (Table 1), and the diameter of raindrops in the study conducted by Ma et al. [12] was 2.4 mm, which was the reason for the difference in aggregate breakdown.

### 4.2. Change in the Mass of Aggregates Transported

Before and after runoff generation, the mass of aggregates under $RI_{78}$ was greater than that under $RI_{127}$ (Table 3). However, the mass of transported aggregates before runoff generation seems to be inconsistent with the conclusion that the splash mass increases with increasing rainfall intensity [8,11]. Therefore, the change in the transport rate of aggregates with rainfall intensity before and after runoff generation was further analyzed (Table 5). The transport rate of aggregates under $RI_{127}$ before runoff generation was 2.33 times that of $RI_{78}$ (Table 5). The results show that the mass of transported aggregates increased with increasing rainfall intensity at the same time. As the impact time of raindrops on soil under $RI_{78}$ was 2.43 times that under $RI_{127}$ (Table 3), the mass of transported aggregates under $RI_{78}$ was higher than that under $RI_{127}$. The transport rate of aggregates under $RI_{78}$ after runoff generation was higher than that under $RI_{127}$, which may be due to the following two reasons: first, the runoff depth and flow velocity under $RI_{127}$ were higher than those under $RI_{78}$ (Table 1); consequently, the loose material in the topsoil was easily removed by runoff, resulting in a decrease in the mass of aggregates that could be transported by raindrops. Second, after runoff generation, the runoff energy under low rainfall intensity was low, which was not sufficient to transport the aggregates out of the soil tank and allow them to deposit at the bottom of the soil tank. Under the impact of raindrops, aggregates would be splashed out. However, the runoff energy under high rainfall intensity was high, which reduced the deposition of aggregates.

**Table 5.** The transport rate of aggregates before and after runoff generation.

| Experimental Treatment | The Transport Rate of Aggregates before Runoff Generation ($mg \cdot cm^{-1} \cdot min^{-1}$) | The Transport Rate of Aggregates after Runoff Generation ($mg \cdot cm^{-1} \cdot min^{-1}$) | The Total Transport Rate of Aggregates under Rainfall ($mg \cdot cm^{-1} \cdot min^{-1}$) |
|---|---|---|---|
| $RI_{78}$ | 3.23 | 5.56 | 2.71 |
| $RI_{127}$ | 7.53 | 2.71 | 3.11 |

Note: $RI_{78}$ and $RI_{127}$ represent rainfall intensities of 78 and 127 $mm \cdot h^{-1}$, respectively.

*4.3. The Variation in the Characteristics of Aggregates with Different Fraction Sizes with the Transport Distance*

With increasing transport distance, the mass percentage of 1–0.053 mm aggregates gradually increased, and that of <0.053 mm aggregates gradually decreased (Table 3). However, Yao et al. [40] considered that the mass percentage of aggregates with a fraction size of 0.02–0.05 mm was the maximum at all transport distances under splashing. The main reason is that the organic matter content of the soil in the study by Yao et al. was low (1.61%), while the organic matter content of the soil in this study was higher (33.7%). Ma et al. [41] stated that soil aggregates with a high organic matter content are more stable; consequently, the macroaggregates in the current study were less likely to break down into microaggregates. In addition, the runoff preferentially transported the microaggregates, which resulted in a lower content of microaggregates splashing out. The changes in the mass percentage of aggregates at different transport distances mainly occurred in the fraction sizes of 1–0.25 and <0.053 mm before and after runoff generation. Compared with the 1–0.25 and <0.053 mm fractions sizes, the 0.25–0.053 mm fractions sizes were most susceptible to transport by thin-layer runoff [42]. The results show that the transport distance of aggregates with fraction sizes of 1–0.25 and <0.053 mm was reduced after runoff generation (Figure 3). The main reason is that the transport effect of raindrops on aggregates was weakened by the influence of thin-layer runoff. The transport distances of the 1–0.25 and <0.053 mm fraction sizes under $RI_{127a}$ were longer than those under $RI_{78a}$ (Figure 3). This difference was mainly because the higher runoff velocity provided a larger lateral force for raindrops, and the raindrop kinetic energy increased in the horizontal direction [43]. Aggregates after runoff generation also had a greater lateral force and could be affected by raindrops over a longer distance.

**5. Conclusions**

In this study (rain intensity: 78,127 $mm \cdot h^{-1}$; slope: 4°), the aggregates transported by rainfall were mainly in the fraction sizes of <1 mm. The total mass percentage of the 1–0.25 and 0.25–0.053 mm fraction sizes after runoff generation increased significantly by 35.9% and 18.9% compared to that before runoff generation under $RI_{78}$ and $RI_{127}$, respectively. The total mass of aggregates transported after runoff generation under $RI_{78}$ was increased significantly by 15.7% compared to that of $RI_{127}$. The mass percentage of 1–0.25 mm aggregates increased by 0.98–1.38 times as the transport distance increased. In general, rainfall evidently affected the 1–0.053 mm fraction size before and after runoff generation, and the mass percentage of macroaggregates increased with increasing transport distance. When runoff was generated, the mass of transported aggregates under low rainfall intensity was greater than that under high rainfall intensity.

The change in soil erosion caused by runoff is a complex dynamic process, and there were still some limitations in this study. Fortunately, we found that the breakdown and transportation characteristics of aggregates were different before and after runoff generation. This provided a basic reference for us to understand the changes in the fraction size distribution of soil aggregates under rainfall conditions. In the future, it is necessary to study the characteristics of aggregate breakdown and transportation before and after runoff generation in different regions or different soil types.

**Author Contributions:** Conceptualization, Y.Z., X.C. and Y.F.; methodology, Y.Z., X.C. and Y.F.; software, Y.Z., H.W. and Y.F.; validation, Y.Z., X.C. and Y.F.; formal analysis, X.C. and Y.F.; investigation, X.C. and Y.F.; resources, X.C. and Y.F.; data curation, Y.Z. and Y.F.; writing—original draft preparation, Y.Z. and Y.F.; writing—review and editing, Y.Z., X.C. and Y.F.; visualization, Y.Z.; supervision, H.W., X.C. and Y.F.; project administration, Y.Z., X.C. and Y.F. All authors have read and agreed to the published version of the manuscript.

**Funding:** This work was supported by the National Key Research and Development Program of China (2021YFD1500705), the Heilongjiang Province Applied Technology Research and Development Program Project (GA20B401), the Fundamental Research Funds for the Central Universities (2572020DR02), China Postdoctoral Science Foundation (2022M710643), the Fundamental Research Funds for the Central Universities (2572021BA05) and Heilongjiang Touyan Innovation Team Program (Technology Development Team for High-Efficiency Silviculture of Forest Resources).

**Institutional Review Board Statement:** Not applicable.

**Informed Consent Statement:** Informed consent was obtained from all subjects involved in the study.

**Data Availability Statement:** All other sources of data are cited throughout the paper.

**Acknowledgments:** We would like to thank the laboratory of soil and water conservation and desertification control of the Forestry College of Northeast Forestry University and the laboratory of Maoershan Experimental Forest Farm in Harbin, Heilongjiang Province, for their technical support.

**Conflicts of Interest:** The authors declare no conflict of interest.

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
