# Peer review of "Effect of Rainfall on Soil Aggregate Breakdown and Transportation on Cultivated Land in the Black Soil Region of Northeast China"

_sustainability, doi:10.3390/su141711028_

Round 1

Reviewer 1 Report

The idea of this paper could be impactful since it relates with soil erosion and nutrient loss management. Hence, this paper could be considered for acceptance after following improvements:

Title:

Title of the paper need to modify

Highlight:

Revised the highlight section

Introduction:

Reference in introduction section need to be updated or delete

 Discussion:

Many sentences repeated the same standpoints that we already know for many years. This part should be focus on discussing the new findings of this research and also provide recent reference

With best regards

Author Response

请参阅附件。

Reviewer 2 Report

The manuscript is dealing with interesting topic of water erosion management. It is a major issue for soil conservation and the findings of this model study can be of importance. However, authors need to highlight the implications of the findings of this study in the abstract and conclusion parts more clearly and in a elaborate manner for getting wider attention of the readers. Overall, content, background literature, methodology and discussion part is well presented. However, authors are advised to highlight the sustainability aspects of the findings at appropriate places in the main text. There are a few minor queries in the methodology and a few suggestions in the other parts of the manuscript which have been annotated in the attached pdf file.

Author Response

请参阅附件。

Reviewer 3 Report

1. A first general remark is that the conclusions are relative to the specific site conditions. It is not clear which general conclusions can be made and can be used in a later similar research. Discussing relative fractions makes a general understanding not clear.

2. The organic matter is very high (+/- 35%).  It is important to make also considerations about the effect of change in organic matter : are there important changes because of erosion/run off : does this affect the soil conditions ? 

3.In the abstract is written that the fraction size 0.25 mm was a critical size. This is not discussed in a clear way in the text. I suppose this conclusion has be seen in relation to figure 3.  But first of all, the horizontal axis has  a logarithmic scale, so defining 0.25 mm as a critical size is not correct. Because of the definition of the enrichment ratio, the 'critical size 0.25 mm' depends on the  original granulometry, so it is not an 'indepedent' conclusion.  

4. Compared with reality, one can expect that, because of the flow of the water and the higher dynamic water pressures on a larger scale,  runoff and erosion will be different  in the test device. Is it possible to discuss more in detail the limitations and the representativity of the test device and the results obtained ?

5. What impact can be expected after several periodes of rain : are the findings, as discussed in the article only valid for the first 'flush' ?

6. In line 171 the turbulent flow is defined as the flow with a Reynolds number > 500. This is not correct. Re = v . D/n  (n = kinematic coefficient of viscosity = 0.01 cm²/s). For a Re = 500 and a mean diameter (for example) 0.1 mm this should result in a velocity 5 m/s.

Author Response

请参阅附件。

Round 2

Reviewer 3 Report

The answers on the questions and remarks are clear to me. It is important to refer to these modifications in your conclusion. They make the 'message' more clear.

Author Response

请参阅附件。
